# Interspecies radiative transition in warm and superdense plasma mixtures

S. X. Hu [1]✉, V. V. Karasiev[1], V. Recoules[2], P. M. Nilson[1], N. Brouwer [2] & M. Torrent [2]

Superdense plasmas widely exist in planetary interiors and astrophysical objects such as brown-dwarf cores and white dwarfs. How atoms behave under such extreme-density conditions is not yet well understood, even in single-species plasmas. Here, we apply thermal density functional theory to investigate the radiation spectra of superdense iron–zinc plasma mixtures at mass densities of $\rho = 250$ to 2000 g cm$^{-3}$ and temperatures of $kT = 50$ to 100 eV, accessible by double-shell–target implosions. Our ab initio calculations reveal two extreme atomic-physics phenomena—firstly, an interspecies radiative transition; and, secondly, the breaking down of the dipole-selection rule for radiative transitions in isolated atoms. Our first-principles calculations predict that for superdense plasma mixtures, both interatomic radiative transitions and dipole-forbidden transitions can become comparable to the normal intra-atomic K$\alpha$-emission signal. These physics phenomena were not previously considered in detail for extreme high-density plasma mixtures at super-high energy densities.

[1] Laboratory for Laser Energetics, University of Rochester, 250 East River Road, Rochester, NY 14623-1299, USA. [2] CEA, DAM, DIF, 91297 Arpajon, France.
✉email: shu@lle.rochester.edu

Extreme material conditions, such as super-high density and warm or hot temperatures, can be widely found in the universe. For example, brown-dwarf cores and white dwarfs[1–3] can have a mass density of $\rho = 10^3$–$10^7$ g cm$^{-3}$ and temperatures up to ~$10^6$ K. Thanks to technological advancements, such extreme states of matter can now be created in the laboratory using powerful lasers[4–7] and/or pulsed-power machines[8]. For instance, deuterium and tritium contained in a millimeter-size inertial confinement fusion (ICF) target can be squeezed to $\rho = 10^2$–$10^3$ g cm$^{-3}$ by powerful lasers through laser-driven compression and spherical convergence[9–13]. Using double-shell implosions[14,15], mid-/high-Z materials can be squeezed to super-high densities ranging from $\rho = 10^3$–$10^4$ g cm$^{-3}$ with a temperature ranging from tens to hundreds of electron volts (1 eV ~ 11,604 K). Understanding how matter behaves at such extreme conditions is the purview of high-energy-density physics, inertial confinement fusion, planetary science, and astrophysics.

Under superdense conditions, atoms and molecules—the fundamental building blocks of matter—can have drastically different properties from those found under ambient conditions. For instance, because of pressure ionization, the binding energy of core electrons of atoms might significantly shift in dense plasmas[16–18] when compared with the case of isolated atoms. By probing the energy level changes in these systems, one can infer the dense-plasma conditions if one knows precisely beforehand how atoms behave in high-density environments. Moreover, such an extreme environment experienced by embedded ions can also alter the characteristics of atomic wavefunctions because of closely encountered neighboring ions. This can have profound implications for understanding radiation transport in such dense plasmas. For example, the dipole-selection rule for isolated atoms can break down in extremely dense plasmas. Most interestingly, if a plasma mixture is compressed to very high densities above $10^3$ g cm$^{-3}$, wave-function overlapping of deeply bound electrons between different atomic species may occur. A schematic diagram of such a scenario is depicted in Fig. 1, in which the iron (Fe) and zinc (Zn) ions in the mixture closely interact with each other in superdense plasma. As a result of the short distance (d) between the two species, their outer electrons on $n = 3$ and $n = 4$ levels can be pressure ionized and their $2s$ and $2p$ states might also be significantly distorted by each other. The significant overlapping of $n = 2$ states could enable a physics phenomenon—interspecies radiative transitions (IRT)—to occur.

As Fig. 1 illustrates, if $1s$ holes of both Fe and Zn ions are created by either radiation pumping[19] or energetic electron collisions[20,21], the $2s$ and $2p$ electrons of one species (e.g., Fe)

could radiatively transition to the $1s$ hole of the other species (e.g., Zn), giving interspecies K$\alpha$ emission. On the other hand, if the $2p$ state is no longer fully occupied and the $1s$-core state is filled, the interspecies K$\alpha$ absorption could occur in such extremely dense-plasma mixtures. To the best of our knowledge, this phenomenon of IRT between bound states has not been considered in emissivity/opacity calculations of plasma mixtures[22–26], even though inter-Coulombic Auger decay was discovered in large molecules and clusters[27–29], and collision-induced absorption and emission between atomic gases was discussed[30–32]. Furthermore, the significant distortion of the $2s$ state resulting from closest neighboring ions will make both intra-atomic and inter-atomic $2s$–$1s$ transitions possible, which are dipole-forbidden for an isolated atom and relatively low-density systems when deeply bound $2s$ and $1s$ states preserve their ideal $s$ symmetry. To classify various transitions, we use the word of intra-atomic for transitions of electron having both initial state and final state belong to the same atom, while inter-atomic transitions involve two atoms that can be either the same type or different species.

Here, we present interspecies radiative transition results from first-principles calculations by thermal density-functional theory (DFT) using the ABINIT software package[33,34] in the plane-wave-based projector augmented-wave (PAW) approach. All electrons are considered as evolving—no frozen core approximation—and spin–orbit coupling effects are explicitly included. As an example for mid-Z elements presented in brown-dwarf cores, a dense-plasma mixture of Fe and Zn was considered with an equal atomic fraction for each species (50:50). We varied the Fe–Zn plasma density from $\rho = 250$ to $2000$ g cm$^{-3}$ and temperatures of kT = 50 to 100 eV. For a chosen plasma condition, we first ran orbital-free DFT-based molecular dynamics[35,36] to obtain the equilibrium ionic configurations. We then took several snapshots of uncorrelated ionic configuration for the electronic structure calculations using ABINIT. Once the electronic structure of a dense plasma is determined from the ABINIT calculations, we created a $1s$-hole state by removing the occupation of the $1s$ state for both Fe and Zn ions. Finally, we calculated the dipole matrices to determine the emission spectra of superdense Fe–Zn plasmas with the Kubo–Greenwood formalism. More numerical details and convergence tests can be found in the "Methods" and Supplementary Information.

## Results

**Interspecies radiative transition in warm and superdense plasmas.** For a superdense and warm Fe–Zn plasma of $\rho = 1000$ g cm$^{-3}$ and kT = 50 eV with $1s$ vacancies of both Fe and Zn ions, the calculated emission coefficient as a function of photon energy

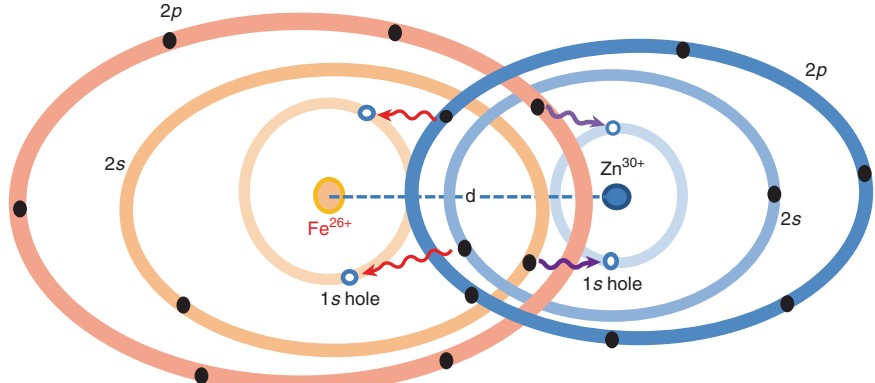

**Fig. 1 The schematic diagram of interspecies radiative transition in superdense plasmas.** Due to the high compression of superdense plasmas, ions of different species can get so close to each other that their atomic states may become overlapping. In such a superdense environment, photon-pumping or collision-induced $1s$-core holes can be filled by $2p$ electrons from other species, giving rise to inter-atomic K$\alpha$ emission.

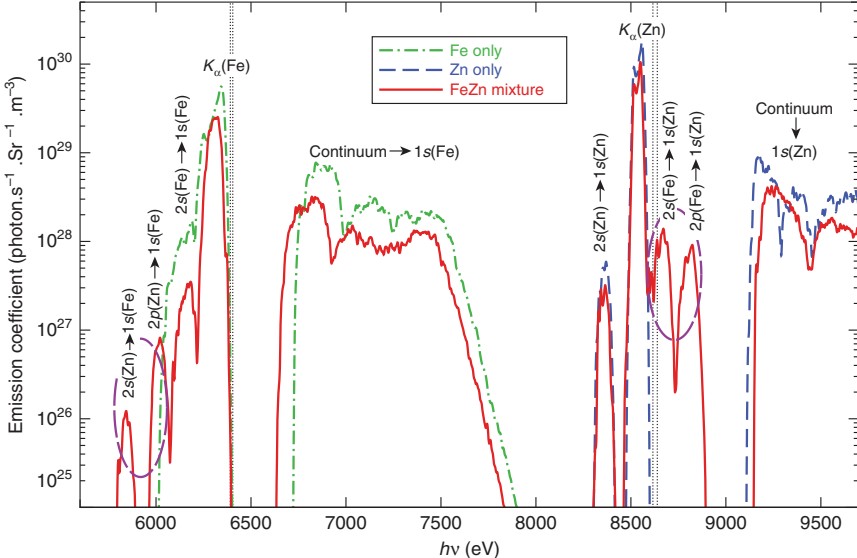

**Fig. 2 The emission spectra of superdense plasmas.** Three cases are calculated for Fe-only, Zn-only, and Fe–Zn mixture having 1s vacancy at $\rho = 1000$ g cm$^{-3}$ and kT = 50 eV, using density-functional theory (DFT) with ABINIT.

is shown by the solid-red line in Fig. 2. To identify the IRT features, we also plotted the spectra of single-species Fe (dashed–dotted green line) and Zn (dashed blue line) plasmas in Fig. 2, respectively. Again, these pure plasmas have the same density and temperature conditions as that of the Fe–Zn mixture.

From Fig. 2, one can clearly see that four additional spectral peaks appear in the superdense Fe–Zn plasma mixtures (highlighted by the dashed ellipse): the two emission lines located at hv ≈ 8666 eV and hv ≈ 8816 eV correspond to transitions from the 2s and 2p states of the Fe ion to the 1s hole of the Zn ion, while the other two peaks at hv ≈ 5838 eV and hv ≈ 6012 eV belong to radiative transitions of 2s/2p electrons of the Zn ion to the 1s vacancy of Fe. Besides these inter-atomic Kα emissions, the dominant intra-atomic Kα lines for each species, are of course, present in the emission spectra in Fig. 2. The vertical dotted black lines mark the normal intra-atomic Kα locations of ambient Fe and Zn, respectively. The red shift of the intra-atomic Kα line is caused by the increased electron screening resulting from the dense plasma environment[19]. In addition, the intra-atomic 2s → 1s transitions for each species, although being about three orders of magnitude weaker than the normal intra-atomic Kα lines, also appear as a consequence of the breaking down of the dipole-selection rule due to the density-induced distortion of 2s states. Finally, the continuum emissions from free electrons filling 1s holes of Fe and Zn ions are also present in the emission spectra, as expected (shown by Fig. 2).

To further understand the emission spectra of Fig. 2, we have computed the density of states (DOS) for the three dense-plasma cases. The results are plotted in Fig. 3, in which Fig. 3a, 3b are for Fe-only and Zn-only plasmas, respectively. One finds that the outer bound states of 3s, 3p, 3d (or 4s) states of Fe and Zn atoms have merged into the continuum because of pressure and thermal ionization. Note that the continuum states below and above the Fermi level (E$_F$) (i.e., chemical potential) are partially occupied. Clearly, the discrete states of 1s, 2s, and 2p of Fe and Zn ions are evidenced in Fig. 3a, 3b. By looking into the occupations on states below the Fermi energy, the estimated average ionizations are <Z> ≈ 17.3 and <Z> ≈ 19.1, respectively, for Fe-only and Zn-only cases. This indicates that the 2p state of Fe begins to be partially occupied. When Fe and Zn plasmas are mixed together, their discrete states of 1s, 2s, and 2p are slightly red/blue shifted by ~15 to 30 eV in Fig. 3c when compared with

the corresponding pure-plasma cases. This shift can be attributed to the interactions between the two species. Now, if their 1s states become empty, i.e. a hole/vacancy is created, the radiative transitions from 2s/2p electrons of Fe and Zn ions to fill 1s holes give rise to the corresponding emission lines in Fig. 2. The breaking down of dipole-selection rule for the 2s → 1s transitions is caused by non-spherical character in the 2s state due to density-induced distortions. Finally, the emission from transitions of continuum to the 1s hole can also be explained.

**Density dependence of interspecies radiative transition (IRT).** To explore how density change affects the interspecies radiative transition in superdense plasmas, we have performed similar first-principles calculations by varying the Fe–Zn density from $\rho = 250$ g cm$^{-3}$ to $\rho = 2000$ g cm$^{-3}$ but keeping kT = 50 eV. The DFT-predicted emission spectra are plotted in Fig. 4 for three different Fe–Zn plasma densities of $\rho = 500$, 1000, and 1500 g cm$^{-3}$. Again, the IRT peaks are highlighted by the dashed ellipses in each panel of the figure.

At a relatively lower density of $\rho = 500$ g cm$^{-3}$, Fig. 4a shows that the inter-atomic Kα emission is significantly weaker than the normal intra-atomic Kα emission by ~3 to 4 orders of magnitude. They are even lower than that of dipole-forbidden intra-atomic 2s → 1s transitions. It is noted that the spin–orbit coupling–induced splitting of Kα1 and Kα2 is clearly seen for Zn, but for Fe they are merged into one peak because of density/temperature broadenings. As the Fe–Zn plasma density increases to $\rho = 1000$ g cm$^{-3}$ and $\rho = 1500$ g cm$^{-3}$, the inter-atomic Kα emission peaks drastically rise in amplitude and their widths increase as a result of strong density broadening (Fig. 4b, c). At an extremely high Fe–Zn density of $\rho = 1500$ g cm$^{-3}$, Fig. 4c indicates that the peak amplitude of inter-atomic Kα from 2p (Fe) → 1s(Zn) transition can approach ~10% of the intra-atomic Kα from 2p(Zn) → 1s(Zn) transition, which should be readily detectable in experiments. It is noted that these inter-atomic transitions become even stronger than the dipole-forbidden intra-atomic 2s(Zn) → 1s(Zn) transition.

One interesting feature seen in Fig. 4 is that the inter-atomic Kα emission from 2p(Fe) → 1s(Zn) transition is always stronger than that of 2p(Zn) → 1s(Fe). To further explore this asymmetry and the overall trend of IRT versus plasma density, we have

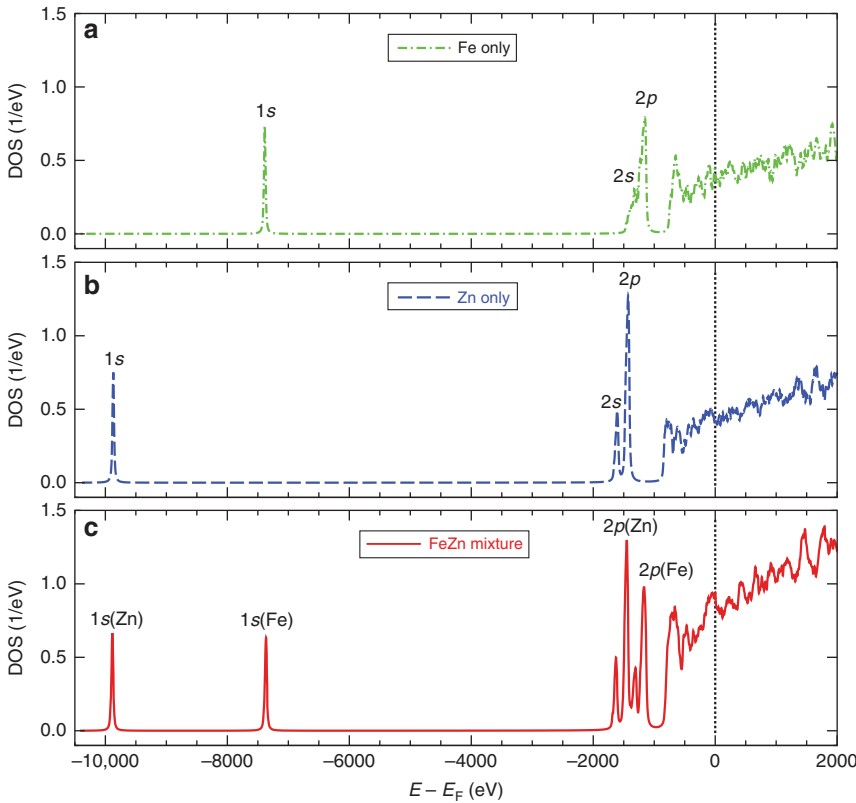

**Fig. 3 The density of states (DOS) of superdense plasmas. a** Pure Fe plasmas, **b** pure Zn plasmas, and **c** Fe–Zn mixture plasmas at the same density and temperature of $\rho = 1000\ \mathrm{g\ cm^{-3}}$ and $kT = 50$ eV.

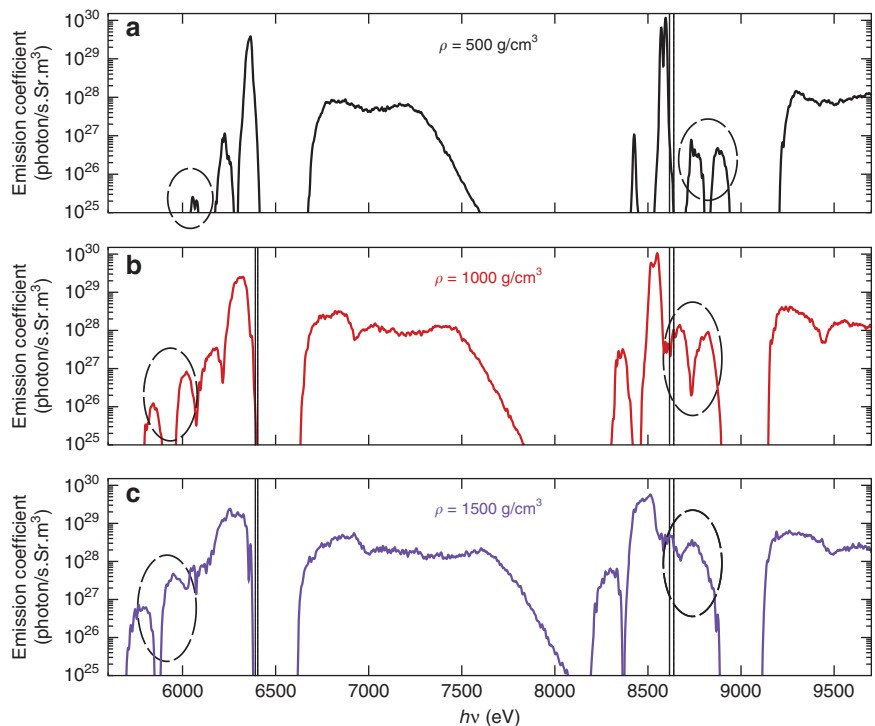

**Fig. 4 Interspecies radiative transitions (IRT) versus plasma density.** DFT calculations are done for superdense Fe–Zn mixtures at $kT = 50$ eV but different plasma mass densities: **a** $\rho = 500\ \mathrm{g\ cm^{-3}}$, **b** $\rho = 1000\ \mathrm{g\ cm^{-3}}$, and **c** $\rho = 1500\ \mathrm{g\ cm^{-3}}$.

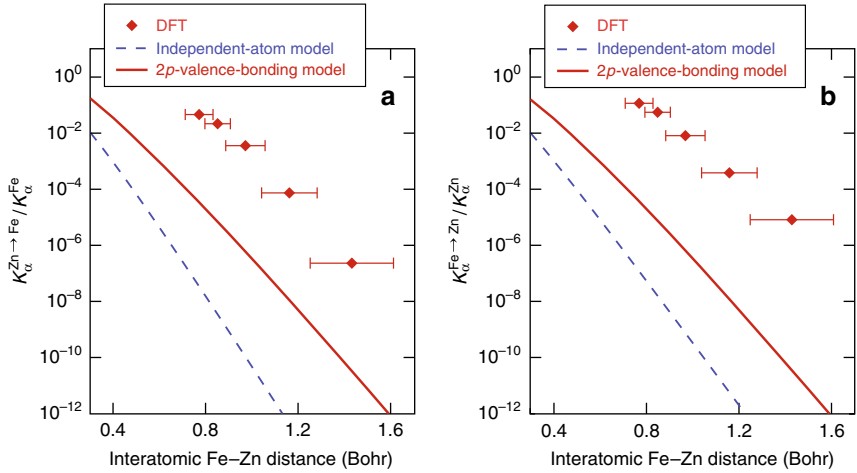

**Fig. 5 Model predictions of IRT in comparison with DFT calculations. a** The ratio of inter-atomic K$\alpha$ emission (from Zn to Fe) to the regular intra-atomic K$\alpha$ signal (within Fe) as a function of inter-atomic Fe–Zn distance. **b** Similar to **a** but for inter-atomic K$\alpha$ emission from Fe to Zn. All of these calculations were done at different plasma densities with the same temperature of kT = 50 eV. DFT stands for density-functional theory. The "error bar" in DFT data represents the full-width half-maximum of g(r) peak.

plotted the DFT-predicted ratio of the inter-atomic K$\alpha$ signal $K_\alpha^{Zn \to Fe}$ or $K_\alpha^{Fe \to Zn}$ to the corresponding intra-atomic emission $K_\alpha^{Fe}$ (or $K_\alpha^{Zn}$) (by the red diamond symbols in Fig. 5) as a function of inter-atomic distance between Fe and Zn ions. For a chosen Fe–Zn plasma density varying from $\rho = 250$ g cm$^{-3}$ to $\rho = 2000$ g cm$^{-3}$ at the same temperature of kT = 50 eV, we derived the Fe–Zn distance (d) from the orbital-free DFT-MD runs, in which d corresponds to the peak location of the pair distribution function—g(r). The full-width half-maximum of g(r) peak gives the plausible range of Fe–Zn distance ("error bar" of d in Fig. 5).

To further see how the inter-atomic K$\alpha$ emission qualitatively changes with d, we have used two simple models to estimate the inter-to-intra K$\alpha$ ratio through the following dipole-matrix elements calculations:

$$\frac{K_\alpha^{Zn(Fe) \to Fe(Zn)}}{K_\alpha^{Fe(Zn)}} = \frac{\left| \int_0^\infty \left\langle \psi_{final}^{model}(\mathbf{r}) | \mathbf{e}.\mathbf{r} | \psi_{initial}^{model}(\mathbf{r} - \mathbf{d}) \right\rangle d\mathbf{r} \right|^2}{\left| \int_0^\infty \left\langle \psi_{1S}^{Fe(Zn)}(\mathbf{r}) | \mathbf{e}.\mathbf{r} | \psi_{2p}^{Fe(Zn)}(\mathbf{r}) \right\rangle d\mathbf{r} \right|^2}, \quad (1)$$

where the unit vector e defines the direction of dipole moment. Depending on how we choose the initial and final states in the nominator of the above equation to estimate the matrix element, we have two simple models: (a) the independent-atom model (IAM) which simply takes the 2p and 1s hydrogenic wavefunctions of independent Fe and Zn atoms as the initial and final states (namely $\left| \psi_{initial}^{IAM}(\mathbf{r}) \right\rangle = \left| \psi_{2p}^{Zn(Fe)}(\mathbf{r}) \right\rangle$ and $\left| \psi_{final}^{IAM}(\mathbf{r}) \right\rangle = \left| \psi_{1s}^{Fe(Zn)}(\mathbf{r}) \right\rangle$.); and (b) the 2p-valence-bonding model in which the initial and final states have a form of symmetrized products of hydrogenic 1s and 2p wavefunctions of both Fe and Zn atoms, that is, $\left| \psi_{initial}^{2p-valence-bonding}(\mathbf{r}_1, \mathbf{r}_2) \right\rangle = \left[ \left| \psi_{2p}^{Fe}(\mathbf{r}_1) \right\rangle \left| \psi_{2p}^{Zn}(\mathbf{r}_2) \right\rangle + \left| \psi_{2p}^{Fe}(\mathbf{r}_2) \right\rangle \left| \psi_{2p}^{Zn}(\mathbf{r}_1) \right\rangle \right]/\sqrt{2}$ and $\left| \psi_{final}^{2p-valence-bonding}(\mathbf{r}_1, \mathbf{r}_2) \right\rangle = \left[ \left| \psi_{1s}^{Fe(Zn)}(\mathbf{r}_1) \right\rangle \left| \psi_{2p}^{Zn(Fe)}(\mathbf{r}_2) \right\rangle + \left| \psi_{1s}^{Fe(Zn)}(\mathbf{r}_2) \right\rangle \left| \psi_{2p}^{Zn(Fe)}(\mathbf{r}_1) \right\rangle \right]/\sqrt{2}$, which is analog to the covalent bonding in molecules. For the latter model, six-dimensional integration over $\mathbf{r}_1$ and $\mathbf{r}_2$ is needed for evaluating the dipole matrix. Using these two models, we can qualitatively estimate the above ratios as a function of the inter-atomic distance d. The results are plotted as the dashed-blue and solid-red lines in Fig. 5a and b,

respectively, for the independent-atom model and the 2p-valence-bonding model. One can see that these simple models qualitatively give the overall trend of increasing inter-atomic K$\alpha$ emission as inter-atomic distance decreasing; Quantitatively, both models show orders of magnitude differences from many-body DFT calculations. Nevertheless, the 2p-valence-bonding model is better than the independent-atom model, which manifests the molecular-bonding nature among atoms in such superdense systems. Molecular bonding involving more than two atoms might account for the discrepancies between the two simple models and DFT calculations. It is noted that the multicenter wave-function nature[37] was properly accounted for by DFT. At the highest density explored ($\rho = 2000$ g cm$^{-3}$), Fig. 5b shows that the inter-atomic $K_\alpha^{Fe \to Zn}$ emission can reach over 11% of the regular intra-atomic $K_\alpha^{Zn}$ signal. The asymmetry that the inter-atomic K$\alpha$ emission from 2p(Zn) → 1s(Fe) transition is always weaker than that of 2p(Fe) → 1s(Zn) is caused by the fact that the 2p state of Fe ion $\left[ \left| \psi_{2p}^{Fe}(\mathbf{r}) > \right] \right.$ spreads much more than $\left| \psi_{2p}^{Zn}(\mathbf{r}) > \right.$ (because the former is less bounded) so that it can have significant overlap with the 1s hole of Zn (see Fig. 1). This consideration is further confirmed by looking into the orbital (wave-function) overlap in such superdense situations.

Finally, we shall discuss the radiative-to-nonradiative decay branching ratio for 1s-core-hole states created in such superdense-plasma mixtures. We shall point out that for super-dense plasmas considered here, the nonradiative Auger decay channel is hard to measure because Auger electrons will quickly thermalize inside the superdense plasma; On the other hand, the radiative decay can be easily probed by measuring the escaped K$\alpha$ photons through spectrometers. To calculate the ratio of K$\alpha$ emission to Auger decay, we have used the atomic kinetic modeling code PrimSPECT[38], which is extensively used in the plasma physics community. For the concerned plasma densities varying from 250 g cm$^{-3}$ to 2000 g cm$^{-3}$ and kT = 50 eV, the averaged ionizations of Fe and Zn ions are about <Z> = 15.8~16.2 and <Z> = 19.5~20.1, respectively. Namely, such superdense plasmas mainly consist of neon-like ions of Fe$^{16+}$ and Zn$^{20+}$, which both have the dominant electronic configuration of 1s$^2$2s$^2$2p$^6$ that is close to what is shown by Fig. 3. Now, if external radiative/collisional pump creates 1s-core-hole state

($1s^1 2s^2 2p^6$) of both ions, we can use PrimSPECT to compute the decay rate coefficients. The calculations give a decay rate of $\Gamma_{rad} = 5.4 \times 10^{14}\,s^{-1}$ for the radiative channel of $1s^1 2s^2 2p^6 \rightarrow 1s^2 2s^2 2p^5$ ($K_\alpha$ emission) for $Fe^{17+}$ ions, while its Auger decay rate is about $\Gamma_{Auger} = 9.8 \times 10^{14}\,s^{-1}$ for the dominant transition of $1s^1 2s^2 2p^6 \rightarrow 1s^2 2s^2 2p^4$. Thus, the radiative-to-Auger branching ratio for $Fe^{17+}$ ions is about $\Gamma_{rad}/\Gamma_{Auger} \approx 0.55$. For $Zn^{21+}$ ions, PrimSPECT calculations give the two decay rate coefficients of $\Gamma_{rad} = 1.01 \times 10^{15}\,s^{-1}$ and $\Gamma_{Auger} = 1.07 \times 10^{15}\,s^{-1}$, respectively, which results in a branching ratio of $\Gamma_{rad}/\Gamma_{Auger} \approx 0.94$. These calculations indicate that the radiative decay channel has the same order of probability as the nonradiative Auger decay for intra-atomic transitions. In other words, one third of $Fe^{17+}$ core-hole ions will decay radiatively, while one half of $Zn^{21+}$ core-hole ions will emit $K_\alpha$ photons through intra-atomic transitions. Given the same physics nature of radiative versus nonradiative decay for both intra-atomic and inter-atomic transitions, we expect the similar branching ratio should hold between the inter-atomic radiative transition and the inter-atomic Coulombic decay[39–41]. Once again, the inter-atomic Coulombic decays[39–41] certainly occur within such superdense plasmas, although they may not be measured as easily as the inter-atomic radiative transitions.

**Possible experiments on inter-atomic Kα emissions.** Experimental verification of these first-principles predictions of inter-atomic $K_\alpha$ emissions can possibly be conducted at the Omega Laser Facility utilizing the platform of double-shell implosions[14,15,42,43]. In a double-shell target, the inner metal shell can be made of mid-$Z$ Fe–Cu or Fe–Zn alloys with a core of D2-gas fill. When a low-$Z$ outer shell (beryllium or polystyrene) is driven symmetrically by the 60-beam OMEGA laser to spherically impact on the inner shell, it can cause the inner Fe–Zn (or Fe–Cu) shell to implode. A small convergence ratio of $C_R = R_{initial}/R_{final} \sim 8$–$10$ of the metal shell could give rise to a mass density of $\rho = 500$ to $2000\,g\,cm^{-3}$ for the inner Fe–Zn (or Fe–Cu) shell at its stagnation[42,43]. To create the $1s$ holes of Fe and Zn/Cu ions, one option is to use the high-intensity OMEGA EP beam to generate MeV electrons that can remove some of the $1s$ electrons of Fe and Zn/Cu ions through collisions. The other option is to fill the double-shell target with mid-$Z$ gases, such as Ar and Kr. As a result, the hot-spot self-emission with a certain amount of hard x-rays could ionize the $1s$ electrons of Fe and Zn/Cu ions by radiation pumping. The latter method has been successfully demonstrated in single-shell implosions on OMEGA. In both ways, the created hollow Fe/Zn/Cu ions in such extremely dense plasmas will give rise to inter-atomic $K_\alpha$ emissions, as we have predicted here. These inter-atomic $K_\alpha$ emissions can be measured by spectrometers with a dynamic range of 100 to 1000.

## Discussion

The two phenomena predicted from our first-principles DFT calculations, which are the interspecies radiative transition and the breaking down of dipole-selection rule in extremely dense-plasma mixtures, can have significant implications to high-energy-density (HED) science, ICF, and astrophysics. For plasma opacity/emissivity calculations, the cross-talk between different species and dipole-forbidden transitions have generally been ignored so far by the HED science community. Our first-principles results show that these inter-atomic radiative transitions can become significant and even comparable with normal intra-atomic transitions. The overall trend of IRT can be qualitatively understood by the independent atom model and the 2p-valence-bonding model; while transient multi-atom molecular

bonding could account for the enhancement of IRT in superdense plasmas. One would expect that these emission/absorption channels, opened up in the warm and extremely dense regime, could affect the radiation transport in ICF (e.g., double-shell targets) and astrophysical objects such as brown-dwarf cores. It is noted that the inter-atomic radiative transitions shall occur in superdense single-species plasmas, although they might be indistinguishable to the normal intra-atomic transitions.

## Methods

Our DFT calculations were performed with the ABINIT software package[33,34], in which electrons are treated quantum-mechanically with a plane-wave finite-temperature Kohn-Sham DFT description. The electrons and ions are in thermodynamic equilibrium with an equal temperature ($T_e = T_i$). The electron–nucleus interaction is described in the PAW approach by a pseudopotential generated with a very small matching radius (rc = 0.2 bohr). All electronic wavefunctions are explicitly computed in the thermal DFT formalism. For the electronic exchange and correlation interactions, we use the generalized gradient approximation (GGA) with the Perdew–Burke–Ernzerhof (PBE) functional[44]. It is noted that the PBE functional has been widely used in DFT calculations for warm-/hot-dense plasmas[12,45–47] that showed good agreements with HED experiments; Our results presented here are insensitive to the choice of exchange-correlation functional, for which the local-density approximation (LDA) gives essentially identical results, except for small energy shifts (see Supplementary Information). To sample the dense-plasma configurations, we have conducted molecular-dynamics simulations based on orbital-free DFT. Namely, under the Born–Oppenheimer approximation, the self-consistent electron density is first determined for an ion configuration. Then, the classical ions are moved by the combined electronic and ionic forces, using Newton's equation. This molecular-dynamics procedure is repeated for thousands of time steps, from which optical property (X-ray emission/absorption) can be directly evaluated. Note that we have applied the periodic boundary condition to our first-principles calculations, with a box size determined by the Fe–Zn density and the number of atoms used. Convergent results for $K_\alpha$ emissions were reached by using 32 atoms in a super cell, the Balderschi mean value point for the Brillouin zone sampling[48], and the highest plane-wave energy cutoff of Ecut ≈68 keV. This high-energy cutoff is necessary to accurately sample the deeply bound 1s-core electrons. Detailed convergence tests can be found in the Supplementary Information.

After we ran the calculations for thousands of OFMD steps, we obtained a sufficiently long trajectory of ionic configurations. We then chose several uncorrelated snapshots from these ionic configurations to calculate the X-ray emission spectra of dense Fe–Zn plasmas by using the Kubo–Greenwood formalism[49,50]. Because of the underestimated bandgap by the PBE functional due to electron self-interaction, the resulting spectra were shifted by a constant of $\delta\omega$ ≈110 eV (~1.5% of the 1s–2p bandgap) to match the $K_\alpha$ locations of ambient Fe and Zn. This is justified by comparing the Hartree–Fock calculated energy 1s–2p gap with the PBE-DFT results. The similar matching technique has shown to work well for the measured $K_\alpha$ emission in warm dense Cu experiments on OMEGA EP.

In the $K_\alpha$-emission calculations, the dipole approximation has been invoked. For the concerned photon energy range of hv = 6.0–8.8 keV, the corresponding electromagnetic waves have wavelengths of $\lambda \approx 1.4$–2.1 Å (2.6–3.97 Bohr). Taking an isolated Fe atom as an example, Hartree–Fock calculations give a size of 2s and 2p states ($\langle 2s\,|r|\,2s\rangle$ or $\langle 2p\,|r|\,2p\rangle$) about ~0.12–0.14 Å, which is one order of magnitude smaller than the wavelength of $K_\alpha$ emissions so that the dipole approximation holds well for intra-atomic transitions. For inter-atomic $K_\alpha$ emissions in superdense Fe–Zn plasmas ($\rho \geq 1000\,g\,cm^{-3}$) concerned here, the inter-atomic Fe–Zn distance is around d = 0.8–1.0 Bohr. Taking this emitting entity of Fe–Zn as a whole, its size is still about ~3–5 times smaller than the $K_\alpha$ wavelength. Nevertheless, this prompts us to consider high-order contributions such as the electric quadrupole emission, which is examined by computing the contribution of electric quadrupole term with the independent-atom model for different densities (i.e., different inter-atomic Fe–Zn distances). The results indicated that the relative contribution ratio of quadrupole to dipole is overall less than ~3.2% (see Supplementary Information).

## Data availability

The data that support the findings of this study are available from the corresponding author upon request. They can be immediately shared through email or any other file-sharing systems.

## Code availability

The codes for $K_\alpha$-emission calculations and IRT models are available from the corresponding author upon request.

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

## Acknowledgements

This material is based upon work supported by the US Department of Energy National Nuclear Security Administration under Award Number DE-NA0003856, the University of Rochester, and the New York State Energy Research and Development Authority. This work is partially supported by US National Science Foundation PHY Grant No. 1802964 for S.X.H. and V.V.K. This report was prepared as an account of work (for S.X.H., V.V. K., and P.M.N.) sponsored by an agency of the U.S. Government. Neither the U.S. Government nor any agency thereof, nor any of their employees, makes any warranty, express or implied, or assumes any legal liability or responsibility for the accuracy, completeness, or usefulness of any information, apparatus, product, or process disclosed, or represents that its use would not infringe privately owned rights. Reference herein to any specific commercial product, process, or service by trade name, trademark, manufacturer, or otherwise does not necessarily constitute or imply its endorsement, recommendation, or favoring by the U.S. Government or any agency thereof. The views and opinions of authors expressed herein do not necessarily state or reflect those of the U.S. Government or any agency thereof.

## Author contributions

S.X.H. conceived the project, performed the DFT calculations, and wrote the initial paper. V.V.K. created the all-electron PAW pseudopotentials and modified the ABINIT absorption code to calculate emission spectra. V.R., N.B., and M.T. have implemented the spin-orbital calculation for optical properties in ABINIT, and provided the code and help on running it. P.M.N. and S.X.H. discussed the possible experimental designs. All authors discussed the results and revised the paper.

## Competing interests

The authors declare no competing interests.
