## [Peer Review File · Nature Communications]

Reviewers' comments:

Reviewer #1 (Remarks to the Author):

The manuscript describes interatomic radiative transitions, which appear in extremely dense plasmas, by ab initio calculations and reveals that the interatomic emission can exceed 10% of the normal intra-atomic emission in highly compressed plasmas which can be achieved by a high-power laser system. The findings have a significant impact on the readers in this field and are suitable for the publication in Nature Communications although the following issue should be addressed before the publication.

The usage of "intra-atomic" is sometimes curious especially in the discussion of the breaking down of the dipole-selection rule. The authors mentioned that the breaking down occurs in the "intra-atomic" 2s to 1s transitions. However, the reviewer believes that the interatomic transition also appears in the system of only Fe (or Zn) atoms. Therefore, we cannot distinguish "intra" and "inter" in the 2s to 1s transition under the extremely high-density condition. If the authors rigorously classify the transitions into "intra" and "inter", the description for the classification should be provided in the manuscript; otherwise, the word of "intra-atomic" is misleading to the readers.

Reviewer #2 (Remarks to the Author):

The manuscript by Hu et al. reports theoretical prediction of two new phenomena in dense plasmas: (i) inter-atomic radiative recombination transitions between the outer shell of one ion and the core vacancy of another and (ii) breakdown of the dipole selection rules for atoms in superdense environment. I am not an expert on plasma physics and can only assess the AMO aspect of the work. Within this narrow view the following questions arise:

1. The basic parameter characterizing a core vacancy is the relative contribution of the radiative and the non-radiative (Auger) decay to the core hole lifetime. Given the pioneering nature of the work, I would expect this to be explored as a function of the density, prior to the discussion of the more detailed questions of inter-atomic and non-dipole radiative transitions.
2. Immediately following from (1) the role of inter-atomic Coulombic decay is of interest as well.
3. Why do the authors expect the dipole approximation to hold under the assumed conditions? What is the relative contribution of, say, electric quadrupole term and how does it depend on the density?
4. The authors employ DFT, routinely referring to it as to first-principles theory. Leaving aside the semantics, DFT does necessarily have an empirical element: the exchange-correlation functional (here - PBE [42]). Why is this particular functional expected to provide meaningful results for the assumed conditions? Has it been ever applied to atoms in high-density environments? Has it ever been benchmarked on anything similar to the systems considered in the present work? Finally, how much do the predictions of the authors depend on the choice of the functional? These are crucial questions on which the predictive power and scientific value of the presented calculations hang. The validity of the chosen approach needs to be firmly established, especially in view of the novelty of the problem.

In summary, the work is potentially of high interest and importance, but in its current form lacks systematic approach to the problem and justification of the chosen theoretical tool. Once these weak points are dealt with satisfactorily, I would be glad to recommend the manuscript for publication in Nature Communications.

Vitali Averbukh

Response to Reviewers' Comments on NCOMMS-19-38932

We thank both reviewers for their professional evaluations on our manuscript. We find these comments and suggestions are very helpful for us to improve the clarity of our manuscript. We shall address the concerns point-by-point by following reviewers' comments in below:

Response to Reviewer #1's Comments

We thank the reviewer for carefully evaluating our manuscript. He/she found "*the findings have a significant impact on the readers in this field and are suitable for the publication in Nature Communications...*" The reviewer raised the following issue for us to address:

Issue-1: "*The usage of "intra-atomic" is sometimes curious especially in the discussion of the breaking down of the dipole-selection rule. The authors mentioned that the breaking down occurs in the "intra-atomic" 2s to 1s transitions. However, the reviewer believes that the interatomic transition also appears in the system of only Fe (or Zn) atoms. Therefore, we cannot distinguish "intra" and "inter" in the 2s to 1s transition under the extremely high-density condition. If the authors rigorously classify the transitions into "intra" and "inter", the description for the classification should be provided in the manuscript; otherwise, the word of "intra-atomic" is misleading to the readers.*"

Response/Action: The reviewer is absolutely correct about the confusion caused by using the word of "intra-atomic" in the original manuscript, without a clear description of the classification between "intra-atomic" and "inter-atomic" transitions. To clarify this, we have added the following sentence on page 5. It reads as "To classify various transitions, we use the word of "intra-atomic" for transitions of electron having both initial state and final state belong to the same atom; while "inter-atomic" transitions involve two atoms that can be either the same type or different species."

Response to Reviewer #2's Comments

We thank the reviewer for providing very important insights of AMO physics to the subject at hand. The concerns raised by the reviewer are legitimate and constructive, which we find very useful for improving our manuscript. We shall carefully address each of the reviewer's points in below, with additional analyses and calculations:

Point 1: “*The basic parameter characterizing a core vacancy is the relative contribution of the radiative and the non-radiative (Auger) decay to the core hole lifetime. Given the pioneering nature of the work, I would expect this to be explored as a function of the density, prior to the discussion of the more detailed questions of inter-atomic and non-dipole radiative transitions.*”

Response: We fully agree with the reviewer on the importance of discussing the branching ratio of radiative to non-radiative (Auger) decays for 1s-core-hole states concerned here. In general, for low-Z plasmas ($Z < 10$) the 1s-hole will be filled more probably by Auger decay than the spontaneous photo-emission channel; While for mid-Z plasmas ($Z = 20 \sim 36$) the two branches of decay channels can become comparable, as what are shown from atomic transition calculations given in below. For dense plasmas that we consider in this manuscript, the Auger decay channel is, however, hard to measure because Auger electrons will quickly thermalize inside the dense plasma. In addition, any sheath fields associated with such HED plasmas can make direct Auger electron measurements infeasible if not impossible. On the other hand, the radiative decay can be easily probed by measuring the escaped K_α photons through spectrometers.

Action: To calculate the radiative-to-nonradiative decay branching ratio, we have used the atomic kinetic modeling code PrimSPECT for the Fe-Zn plasma conditions concerned. For the plasma densities varying from 250-g/cm^3 to 2000-g/cm^3 and $kT = 50\text{-eV}$, the averaged ionizations of Fe and Zn ions are about $\langle Z \rangle = 15.8 \sim 16.2$ and $\langle Z \rangle = 19.5 \sim 20.1$, respectively. Namely, such super-dense plasmas mainly consist of neon-like ions of Fe^{16+} and Zn^{20+} , which both have the dominant electronic configuration of $1s^2 2s^2 2p^6$ that is close to what is shown by Fig. 3 in the original manuscript. Now, if external radiative/collisional pump creates 1s-core hole state ($1s^1 2s^2 2p^6$) of both ions, we can use PrimSPECT to compute the decay rate coefficients. The calculations give a decay rate of $\Gamma_{rad} = 5.4 \times 10^{14} \text{ s}^{-1}$ for the radiative channel of $1s^1 2s^2 2p^6 \rightarrow 1s^2 2s^2 2p^5$ (K_α emission) for Fe^{17+} ions, while its Auger decay rate is about $\Gamma_{Auger} = 9.8 \times 10^{14} \text{ s}^{-1}$ for the dominant channel of $1s^1 2s^2 2p^6 \rightarrow 1s^2 2s^2 2p^4$. Thus, the radiative-to-Augur branching ratio for Fe^{17+} ions is about $\Gamma_{rad}/\Gamma_{Auger} \approx 0.55$. For Zn^{21+} ions, PrimSPECT calculations give the two decay rate coefficients of $\Gamma_{rad} = 1.01 \times 10^{15} \text{ s}^{-1}$ and $\Gamma_{Auger} = 1.07 \times 10^{15} \text{ s}^{-1}$, respectively, which results in a branching ratio of $\Gamma_{rad}/\Gamma_{Auger} \approx 0.94$. These calculations indicate that the radiative decay channel concerned in our study here has the same order of probability as the nonradiative Auger decay for intra-atomic transitions. In other words, one third of Fe^{17+} core-hole ions will decay radiatively, while one half of Zn^{21+} core-hole ions will emit K_α photons through intra-atomic transitions. Given the same physics nature of radiative *versus* nonradiative decay for both intra-atomic and inter-atomic transitions, we expect the similar branching ratio should hold between the inter-atomic radiative transition and the inter-atomic Coulombic decay. Once again, the inter-atomic Coulombic decays certainly occur within such super-dense plasmas, although they may not be measured as easily as the inter-atomic radiative transitions.

Revision to main text: To share these important discussions with readers, we have added the following paragraph on pages 13 and 14: “Finally, we shall discuss the radiative-to-nonradiative decay branching ratio for 1s-core-hole states created in such super-dense plasma mixtures. We shall point out that for super-dense plasmas considered here, the nonradiative Auger decay channel is hard to measure because Auger electrons will quickly thermalize inside the super-dense plasma; On the other hand, the radiative decay can be easily probed by measuring the escaped $K\alpha$ photons through spectrometers. To calculate the ratio of $K\alpha$ emission to Auger decay, we have used the atomic kinetic modeling code PrimSPECT³⁸ which is extensively used in the plasma physics community. For the concerned plasma densities varying from 250-g/cm³ to 2000-g/cm³ and $kT=50$ -eV, the averaged ionizations of Fe and Zn ions are about $\langle Z \rangle=15.8\sim 16.2$ and $\langle Z \rangle=19.5\sim 20.1$, respectively. Namely, such super-dense plasmas mainly consist of neon-like ions of Fe^{16+} and Zn^{20+} , which both have the dominant electronic configuration of $1s^2 2s^2 2p^6$ that is close to what is shown by Fig. 3. Now, if external radiative/collisional pump creates 1s-core-hole state ($1s^1 2s^2 2p^6$) of both ions, we can use PrimSPECT to compute the decay rate coefficients. The calculations give a decay rate of $\Gamma_{rad} = 5.4 \times 10^{14} s^{-1}$ for the radiative channel of $1s^1 2s^2 2p^6 \rightarrow 1s^2 2s^2 2p^5$ ($K\alpha$ emission) for Fe^{17+} ions, while its Auger decay rate is about $\Gamma_{Auger} = 9.8 \times 10^{14} s^{-1}$ for the dominant transition of $1s^1 2s^2 2p^6 \rightarrow 1s^2 2s^2 2p^4$. Thus, the radiative-to-Auger branching ratio for Fe^{17+} ions is about $\Gamma_{rad}/\Gamma_{Auger} \approx 0.55$. For Zn^{21+} ions, PrimSPECT calculations give the two decay rate coefficients of $\Gamma_{rad} = 1.01 \times 10^{15} s^{-1}$ and $\Gamma_{Auger} = 1.07 \times 10^{15} s^{-1}$, respectively, which results in a branching ratio of $\Gamma_{rad}/\Gamma_{Auger} \approx 0.94$. These calculations indicate that the radiative decay channel has the same order of probability as the nonradiative Auger decay for intra-atomic transitions. In other words, one third of Fe^{17+} core-hole ions will decay radiatively, while one half of Zn^{21+} core-hole ions will emit $K\alpha$ photons through intra-atomic transitions. Given the same physics nature of radiative *versus* nonradiative decay for both intra-atomic and inter-atomic transitions, we expect the similar branching ratio should hold between the inter-atomic radiative transition and the inter-atomic Coulombic decay³⁹⁻⁴¹. Once again, the inter-atomic Coulombic decays³⁹⁻⁴¹ certainly occur within such super-dense plasmas, although they may not be measured as easily as the inter-atomic radiative transitions.” The following new and relevant references are accordingly added:

[38] MacFarlane, J. J., Golovkin, I. E. *et al.* Simulation of the ionization dynamics of aluminum irradiated by intense short-pulse lasers, in: Proceedings of Inertial Fusion and Science Applications 2003, American Nuclear Society, LaGrange Park IL, 2004.

[39] Averbukh, V. & Cederbaum, L. S. Interatomic electronic decay in endohedral fullerenes. *Phys. Rev. Lett.* **96**, 053401 (2006).

[40] Averbukh, V., Saalman, U., Rost, J. M. Suppression of Exponential Electronic Decay in a Charged Environment. *Phys. Rev. Lett.* **104**, 233002 (2010).

[41] Cooper, B. & Averbukh, V. Single-Photon Laser-Enabled Auger Spectroscopy for Measuring Attosecond Electron-Hole Dynamics. *Phys. Rev. Lett.* **111**, 083004 (2013).

Point 2: “Immediately following from (1) the role of inter-atomic Coulombic decay is of interest as well.”

Response/Action: See our response/action to the above point 1.

Point 3: “Why do the authors expect the dipole approximation to hold under the assumed conditions? What is the relative contribution of, say, electric quadrupole term and how does it depend on the density?”

Response: For the concerned photon energies of K_{α} emissions ($h\nu=6.0\sim 8.8$ keV), the corresponding wavelengths range from $\lambda \approx 1.4 \sim 2.1$ Å (2.6~3.97 Bohr). Taking an isolated Fe atom as an example, Hartree-Fock calculations give a size of $2s$ and $2p$ states ($\langle 2s|r|2s\rangle$ or $\langle 2p|r|2p\rangle$) about $\sim 0.12\text{-}0.14$ Å, which is one order of magnitude smaller than the wavelength of K_{α} emissions so that the dipole approximation holds well for intra-atomic transitions. For inter-atomic K_{α} emissions in super-dense Fe-Zn plasmas ($\rho \geq 1000\text{-g/cm}^3$) concerned here, the inter-atomic Fe-Zn distance is around $d=0.8\sim 1.0$ Bohr (see Fig. 5). Taking this “emitting entity” of Fe-Zn as a whole, its size is still about $\sim 3\text{-}5$ times smaller than the K_{α} wavelength. Nevertheless, this might prompt the concern about high-order contributions such as the electric quadrupole emission pointed out by the referee. Exactly following what the referee suggested, we have used the independent-atom model to examine the relative contribution of electric quadrupole term by comparing it to the dominant dipole contribution, for different densities (*i.e.*, different inter-atomic Fe-Zn distances). The results are shown by the following figure, in which we plot the contribution ratio of quadrupole to dipole terms as a function of inter-atomic distance:

One clearly sees that as the inter-atomic distance increases the “emitting entity” gets bigger so that the quadrupole contribution rises. However, for the density range concerned, the maximum quadrupole term only gives less than $\sim 3.2\%$ of the dipole contribution!

Action: To share these discussions with readers, we have added the following paragraph in the section of Methods of the main text: “In the K_{α} -emission calculations, the dipole approximation has been invoked. For the concerned photon energy range of $h\nu=6.0\sim 8.8$ keV, the corresponding electromagnetic waves have wavelengths of $\lambda \approx 1.4 \sim 2.1$ Å (2.6~3.97 Bohr). Taking an isolated Fe atom as an example, Hartree-Fock calculations give a size of $2s$ and $2p$ states ($\langle 2s|r|2s\rangle$ or $\langle 2p|r|2p\rangle$) about $\sim 0.12\text{-}0.14$ Å, which is one order of magnitude smaller than the wavelength of K_{α} emissions so that the dipole approximation holds well for intra-atomic transitions. For inter-atomic K_{α} emissions in super-dense Fe-Zn plasmas ($\rho \geq 1000\text{-g/cm}^3$) concerned here, the inter-atomic Fe-Zn distance is around $d=0.8\sim 1.0$ Bohr. Taking this “emitting entity” of Fe-Zn as a whole, its size is still about $\sim 3\text{-}5$ times smaller than the K_{α} wavelength. Nevertheless, this prompts us to consider high-order contributions such as the electric quadrupole emission, which is examined by computing the contribution of electric quadrupole term with the independent-atom model for different densities (*i.e.*, different inter-atomic Fe-Zn distances). The results indicated that the relative contribution ratio of quadrupole to dipole is overall less than $\sim 3.2\%$ (see Supplementary Material).”

Moreover, the following details are now given in the Supplementary Material: “To examine the high-order electric quadrupole contribution to inter-atomic K_{α} emissions, we have computed its relative amplitude to that of dipole contribution with the independent-atom model. In Fig. S6, we plot the ratio of quadrupole to dipole contributions as a function of inter-atomic Fe-Zn distance:

Fig. S6 The ratio of quadrupole to dipole contributions as a function of inter-atomic Fe-Zn distance.

One clearly sees that as the inter-atomic distance increases the “emitting entity” gets bigger so that the quadrupole contribution rises. However, for the density range concerned, the maximum quadrupole term only gives less than $\sim 3.2\%$ of dipole contribution!”

Point 4: “The authors employ DFT, routinely referring to it as to first-principles theory. Leaving aside the semantics, DFT does necessarily have an empirical element: the exchange-correlation functional (here - PBE [42]). Why is this particular functional expected to provide meaningful

results for the assumed conditions? Has it been ever applied to atoms in high-density environments? Has it ever been benchmarked on anything similar to the systems considered in the present work? Finally, how much do the predictions of the authors depend on the choice of the functional? These are crucial questions on which the predictive power and scientific value of the presented calculations hang. The validity of the chosen approach needs to be firmly established, especially in view of the novelty of the problem.”

Response: We thank the reviewer for raising this important question on the validity of density-functional theory (DFT) for warm and dense plasma conditions. DFT is a mean-field theory for quantum many-body systems, which is widely used for condense matter physics and quantum chemistry. Yes, the reviewer is absolutely right that the “*holy grail*” of DFT is the exchange-correlation (xc) functional. It is often derived from free electron gas calculations with different levels of accuracy such as local-density approximation (LDA), generalized gradient-density approximation (GGA) [PBE as one of its type], meta-GGA, hybrid, and so on (by climbing the so-called “Jacob’s ladder”). The sophistication of xc-functional is often chosen by balancing its accuracy with computational cost.

First of all, to assure the reviewer that our results are insensitive to the choice of PBE, we have performed two additional DFT calculations with both LDA-PZ (Perdew-Zunger) and LDA-PW92 (Perdew-Wang-92). The results for Fe-Zn plasmas of $\rho=1500\text{-g/cm}^3$ and $kT=50\text{-eV}$ are shown in the following figure:

One can see that these results are almost identical, except for a small ($\sim 10\text{-}15$ eV) energy shift with each other. The inter-atomic K_α signals and their amplitude relative to the normal K_α are unchanged. In general, PBE should be more accurate than LDA.

Secondly, the PBE-DFT calculations of K_α emissions in warm-/hot-dense plasmas have been proven to be valid when compared with experiments. In a recent Omega experiment, a solid-density copper foil was irradiated by 10-ps high-power laser pulse of $\sim 900\text{-J}$ energy; the laser

produced \sim MeV electrons heat the solid target, leading to its temperature rising up from room-temperature to over 500-eV. Meanwhile, the MeV electrons also knock out the 1s-core electron of some Cu atoms, giving rise to K_{α} emissions that are recorded by a time-resolved streaking camera. We compare the PBE-DFT calculations with the measurements (left panel) of the spectral centroid as a function of time in the following figure (right panel):

The above figure shows the good agreement between the experimental measurements and PBE-DFT calculations for such warm-/hot-dense plasmas. One more example: in another recent LCLS experiment the K-edge energy of warm-dense Al was measured in comparison with PBE-DFT calculations. The published results also showed very good agreement [S.M. Vinko, O. Ciricosta & J.S. Wark, *Density functional theory calculations of continuum lowering in strongly coupled plasmas*. Nature Communications **5**, 3533 (2014)], which is illustrated by the following figure (taken from the paper):

Figure 4 | Comparison of calculated and experimentally measured K-edge energies in Al. K-edge energies for Al for a range of ion charge states. The band corresponds to the experimentally determined edge positions with errors, as given by ref. 4, plotted alongside the SP, EK and IS models. The results from this work are marked DFT.

Finally, we shall point out that the thermal DFT method has been a powerful *working-horse* for *first-principles* studies of high-energy-density (HED) physics and chemistry. In general, it often provides very good agreement with experimental measurements for warm and dense plasmas. Some recent publications of using DFT to understand HED experiments are listed here (for results from other groups): *C. J. Pickard and R. J. Needs, Nature materials* **9**, 624 (2010); *P. Davis et al., Nat. Communications* **7**, 11189 (2016); *M. D. Knudson and M. P. Desjarlais Phys. Rev. Lett.* **118**, 035501 (2017); *T. Döppner et al., Phys. Rev. Lett.* **121**, 025001 (2018); just name a few.

Action: To share these discussions with readers, we have added the following sentences in the main text and in the Supplementary Material.

- **Changes to main text:** The following two sentences have been added in the Methods section: “It is noted that the PBE functional has been widely used in DFT calculations for warm-/hot-dense plasmas⁴⁷⁻⁵⁰ that showed good agreements with HED experiments; Our results presented here are insensitive to the choice of exchange-correlation functional, for which the local-density approximation (LDA) gives essentially identical results except for small energy shifts (see Supplementary Material).”; with the following new references:

[47] *Pickard, C. J. & Needs, R. J. Aluminium at terapascal pressures. Nature materials* **9**, 624 (2010).

[48] Vinko, S. M., Ciricosta, O. & Wark, J. S. Density functional theory calculations of continuum lowering in strongly coupled plasmas. *Nature Communications* **5**, 3533 (2014).

[49] Davis, P. et al., X-ray scattering measurements of dissociation-induced metallization of dynamically compressed deuterium. *Nat. Communications* **7**, 11189 (2016).

[50] Döppner, T. et al., Absolute Equation-of-State Measurement for Polystyrene from 25 to 60 Mbar Using a Spherically Converging Shock Wave. *Phys. Rev. Lett.* **121**, 025001 (2018).

- **Changes to Supplementary Material:** The following paragraph has been added to Supplementary Material: “To verify if our results are insensitive to the choice of exchange-correlation functionals, we have performed two additional DFT calculations with both LDA-PZ (Perdew-Zunger) and LDA-PW92 (Perdew-Wang-92). The results for Fe-Zn plasmas of $\rho=1500\text{-g/cm}^3$ and $kT=50\text{-eV}$ are shown by Fig. S5:

Fig. S5 Comparison of emission spectra of Fe-Zn plasmas at $\rho=1500\text{ g/cm}^3$ and $kT=50\text{ eV}$ (32-atom super-cell), using different exchange-correlation functionals in *ABINIT* calculations.

One can see that these results are almost identical, except for a small ($\sim 10\text{-}15\text{ eV}$) energy shift with each other. The inter-atomic K_α signals and their amplitude relative to the normal K_α emission are unchanged between LDA and PBE calculations.”

In addition, we have also updated the *independent atom model* and added the *2p-valence-bonding model* to help understanding the DFT results. Accordingly, we modify the text on page 12 to reflect these minor changes.

With these detailed responses, additional calculations and analyses, and according changes made, we hope the reviewers can now recommend our revised manuscript for publication in Nature Communications.

Reviewer #1 (Remarks to the Author):

The manuscript was revised by following the reviews' suggestions and is now acceptable for the publication in Nature Communications.

Reviewer #2 (Remarks to the Author):

In their response and in the revised manuscript, the authors have provided fully satisfactory answers to all the concerns raised in my initial report. I find the additional numerical demonstrations, especially of the stability of the results with respect to the choice of the exchange-correlation functional, impressive. I therefore strongly recommend the work for publication in Nature Communications.

Vitali Averbukh

Response to Reviewers' Comments on NCOMMS-19-38932A

We thank both reviewers for their professional evaluations on our manuscript. We shall respond to the reviewers' final comments in below:

Response to Reviewer #1's Comments

We thank the reviewer for agreeing that the revised manuscript is "*now acceptable for the publication in Nature Communications*".

Response to Reviewer #2's Comments

We thank the reviewer for providing his final comments on our revised manuscript: "*In their response and in the revised manuscript, the authors have provided fully satisfactory answers to all the concerns raised in my initial report. I find the additional numerical demonstrations, especially of the stability of the results with respect to the choice of the exchange-correlation functional, impressive. I therefore strongly recommend the work for publication in Nature Communications.*"

Response: We thank the reviewer for strongly recommending our revised manuscript for publication in Nature Communications.